# Innovation in NBS Co-Design and Implementation

**James M. Strout** [1,*], **Amy M. P. Oen** [1], **Bjørn G. Kalsnes** [1], **Anders Solheim** [1], **Gerd Lupp** [2], **Francesco Pugliese** [3] and **Séverine Bernardie** [4]

1   Norwegian Geotechnical Institute (NGI), 0855 Oslo, Norway; ao@ngi.no (A.M.P.O.); bgk@ngi.no (B.G.K.); as@ngi.no (A.S.)
2   Chair for Strategic Landscape Planning and Management, Technical University of Munich, 85354 Freising, Germany; gerd.lupp@tum.de
3   Department of Civil, Environmental and Architectural Engineering, University of Naples Federico II, Via Claudio 21, 80125 Naples, Italy; francesco.pugliese2@unina.it
4   Ground Instabilities and Erosion Risk Unit, Risks and Prevention Division, Bureau de Recherches Géologiques et Minières (BRGM), 3 av. Claude-Guillemin, P.B. 36009, CEDEX 02, 45060 Orléans, France; s.bernardie@brgm.fr
*   Correspondence: james.michael.strout@ngi.no

**Abstract:** Impacts in the form of innovation and commercialization are essential components of publicly funded research projects. PHUSICOS ("According to nature" in Greek), an EU Horizon 2020 program (H2020) Innovation Action project, aims to demonstrate the use of nature-based solutions (NBS) to mitigate hydrometeorological hazards in rural and mountainous areas. The work program is built around key innovation actions, and each Work Package (WP) leader is specifically responsible for nurturing innovation processes, maintaining market focus, and ensuring relevance for the intended recipients of the project results. Key success criteria for PHUSICOS include up-scaling and mainstream implementation of NBS to achieve broader market access. An innovation strategy and supporting tools for implementing this within PHUSICOS has been developed and key concepts forming the basis for this strategy are presented in this research note.

**Keywords:** innovation; up-scaling; nature-based solutions (NBS); hydrometeorological hazards; PHUSICOS project; flooding; landslides; avalanches; rockfall; Europe





## 1. Introduction

### 1.1. PHUSICOS

PHUSICOS is an H2020 demonstration project (grant agreement no. 776681) focused on the application of nature-based solutions (NBS) to mitigate hydro-meteorological hazards, such as flooding or landslides in rural and mountainous areas. PHUSICOS aims to integrate existing state-of-the-art methods and technologies in practical settings, and over time to develop an evidence base regarding the performance of these solutions. A key aspect of the project is to employ the concepts of co-design (stakeholder involvement) in the implementation of test cases, effectively anchoring the research results as practical and applied solutions relevant for the intended users. A second key aspect is enabling the demonstration and up-scaling of NBS for real-world applications.

PHUSICOS is one of a cluster of projects funded under the H2020 call number SC5-08-2017, including RECONECT (grant agreement no. 776866) and OPERANDUM (grant agreement no. 776848). The common focus of these projects is the demonstration and up-scaling of NBS. Innovation builds on the practical implementation of research results and is a key expectation of this call. Thematically, these projects fall under the priority societal challenge "Protection of the Environment: Sustainable Management of Natural Resources, Water, Biodiversity, and Ecosystems: and the program "Assess Impacts and Vulnerabilities and Develop Innovative, Cost-Effective Adaptation and Risk Prevention

and Management Measures". The NBS technologies to be demonstrated result from years of developmental research across Europe and internationally.

This document is a project technical note exploring the approach to innovation implemented within the PHUSICOS consortium. The underlying aim of the innovation processes in PHUSICOS is to help bring innovation into the mainstream thinking of researchers and to illustrate the opportunities for open innovation and commercialization as alternatives for (or concurrent processes with) traditional dissemination. The purpose of this review is to reflect on innovation opportunities in research, specifically presenting mechanisms and tools used in PHUSICOS to enable (and encourage) innovation.

### 1.2. What Is an Innovation?

Many definitions exist for innovation, whereby the definitions often reflect the context or paradigms of the field where the definition is applied [1]. As a demonstration project to reduce geohazards with NBS, PHUSICOS represents applied research in the context of value creation for stakeholders. One broad definition of innovation seems appropriate for this context: "Innovation is the process of creating value by applying novel solutions to meaningful problems" [2]. In PHUSICOS, we further refine this definition as the successful exploitation of NBS research results to produce tangible benefits, for example satisfying the needs and wants of relevant users. These benefits may be in terms of societal benefits, promoting sustainability and resiliency, improved life quality, and direct economic activity. Innovations may take the form of technology innovations, e.g., techniques, methods, data or other products and services (Figure 1), or service and social innovations closely linked to the technological innovations through the explicit objectives of producing social impact co-benefits [3].

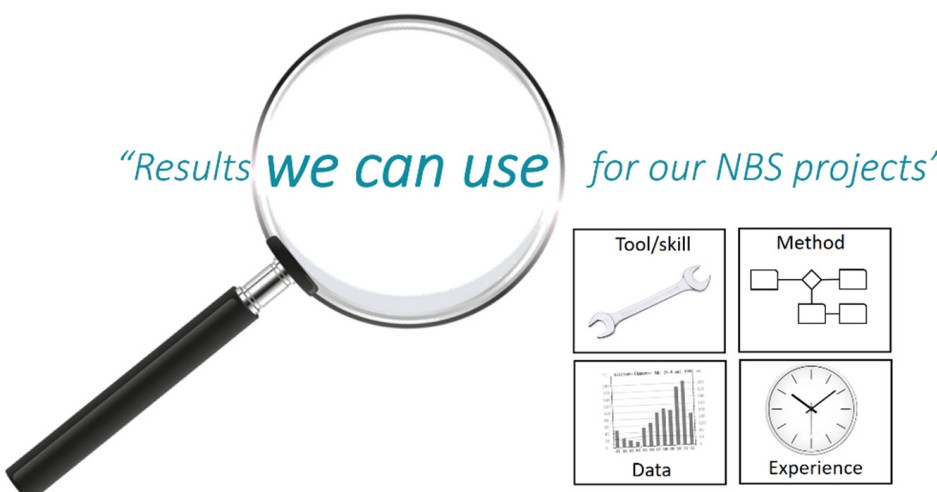

**Figure 1.** Defining technical innovation (illustration by author).

Potential innovations come in all sizes and forms. Most often "showcase innovations" are the culmination of a series of "incremental improvements" [4], e.g., the "showcase" project results are often built on the stepping stones of smaller ones (Figure 2). A potential pitfall is that these incremental innovations may not even be recognized as innovations by the researcher producing them, as from their perspective these innovations may seem minor or inconsequential. However, for others less engaged in the topic they may have great importance. Innovation opportunities may be lost simply because the potential of the results is not recognized or the significance is underrated.

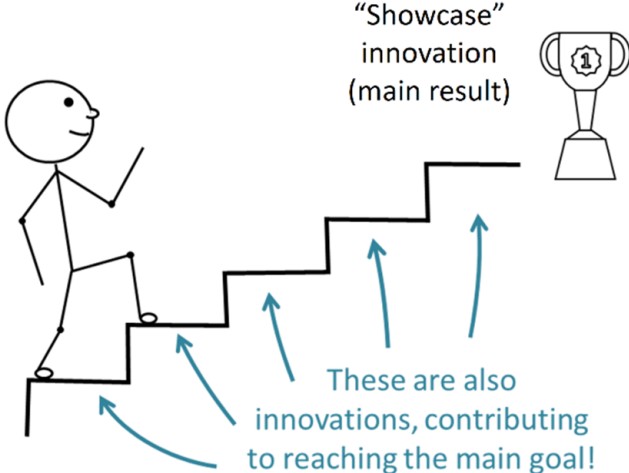

**Figure 2.** Small innovations are stepping stones to larger ones (illustration by author).

Innovations may be technological, social, or related to service; they may also include scaling (up-sizing) of existing solutions, sharing of tools and knowledge across national borders, and converting experiences and know-how into shareable knowledge.

Preserving these opportunities and promoting the innovation related to project results is a specific goal within demonstration projects such as PHUSICOS. Demonstration projects are a form of practical verification of earlier research, and inherently the results from these projects are quite close to market-ready solutions, requiring only market access and the enablement of up-scaling and distribution. Innovative use of these as commercial products or as open access solutions leverages earlier research investments and helps to realize the full potential of these.

### 1.3. Stakeholder Expectations

The innovation process is a value chain [5,6], whereby research results represent the starting point and the culmination of the process is the strategic implementation of these as improved products, services, methods, or knowledge able to create value for the user of the results [7]. Specifically, in PHUSICOS, results from NBS research become innovations when they provide value for a diverse set of stakeholders (adapted from the PHUSICOS Description of Action):

- National, European, and international administrators and policy-makers;
- Local, regional, and national practitioners and entrepreneurs;
- Private sector to include insurance, green banks, and other businesses;
- Environmental groups and other Non-governmental organizations (NGOs);
- Academic networks working with disaster relief reduction, climate adaptation, water management, and the implementation of NBS;
- Site-specific stakeholders from our demonstration case sites, including local citizens;
- Stakeholders from other rural mountain communities, which may benefit from the PHUSICOS demonstration site experiences.

This set represents a broad swath of potential stakeholders. For simplicity, these stakeholders can be grouped to identify their common characteristics and interests (Table 1).

The demonstrator sites are the core of PHUSICOS, and the organization of the project consists of work packages (WPs) building thematically around the cases (demonstrator sites). Stakeholder interests and needs are captured using the living labs (LL) methodology, whereby the application of LL to rural NBS implementations for hazard mitigation may be intended itself as an innovation. The relevance of PHUSICOS within society can be promoted by identifying opportunities to grow NBS technologies for hazard mitigation in business policy forums.

**Table 1.** Stakeholder groups.

| Groups [8] | Characteristics | General Interests |
|---|---|---|
| Commercial sector | Private companies and consultants providing services, such as construction, supply of materials, services, etc. | Providing services and solutions creating value for the customer and the company. This group requires efficiency, quality, risk reduction. |
| Media (information sources and lobbyists) | Media, public interest groups consisting of groups of citizens or organizations dedicated to pushing forward specific interests, needs, or wants (interest groups/advocacy groups/coalition groups). | Promoting information dissemination. Promoting the interests of a segment of society. This group requires information and evidence. |
| Political representatives (authorities) | Government bodies, public agencies or regulatory agencies serving citizens and companies. | Implementing policies and actions to manage, protect, and improve society. |
| Transnational and international organizations | Public organizations operating across national boundaries and often as a collaboration between nations or multi-national private interests. | Promoting social justice and economic equality, enabling development and growth. |
| Academia (experts) | The scientific community. | Research and development to improve knowledge and provide knowledge-based services to other stakeholders. |
| Civil society (citizens) | Individual citizens or persons who have their own personal interests and needs, not belonging within other groups. | Motivations and interests vary. |

Additional stakeholder needs and expectations are met through various forms of innovation:

- Technical (science and engineering of NBS solutions);
- Service (stakeholder involvement through LL);
- Governance (policy and promotion of NBS or sustainable solutions);
- Learning arenas (training and education tools);
- Specific products, software, or knowledge bases.

Finally, the researchers and research organizations have expectations regarding their research results (intellectual property—IP). In the broadest sense, a researcher can choose between publication and dissemination or commercial exploitation. In practice, we may include elements of both.

Publication of research creates provenance (e.g., identifies who created the results), formal IP protection through copyrights, and helps to ensure that the results reach a broader audience and are taken into use, while providing appropriate credit to the researchers and research organizations who produced the results. This is by far the most common choice for most researchers.

Legal protection of the IP may be established, for example patents, copyrights, and trademarks, all of which create legally defined rights for the disposition and exploitation of IP. Exploitation of IP may include economic instruments such as licensing or royalties for access to the IP. The results have the potential to reach a broader audience in the form of commercial products and services.

## 2. Innovation in PHUSICOS

### 2.1. Project-Level Innovation Process

The PHUSICOS project team includes a central technical role to enable and promote innovation and commercialization processes. As a demonstration project, PHUSICOS applies research results to real-world test sites, creating innovation opportunities within the project work packages. The purpose of the innovation manager role is to assist the partners in the process of identifying innovations and bringing these to the market as an alternative to traditional academic dissemination.

The pathway to the market may be through traditional commercialization or via open innovation (Figure 3). Although the figure illustrates individual paths, in practice there will be interaction and cross-linking in terms of dissemination activities related to both open innovation and commercial exploitation.

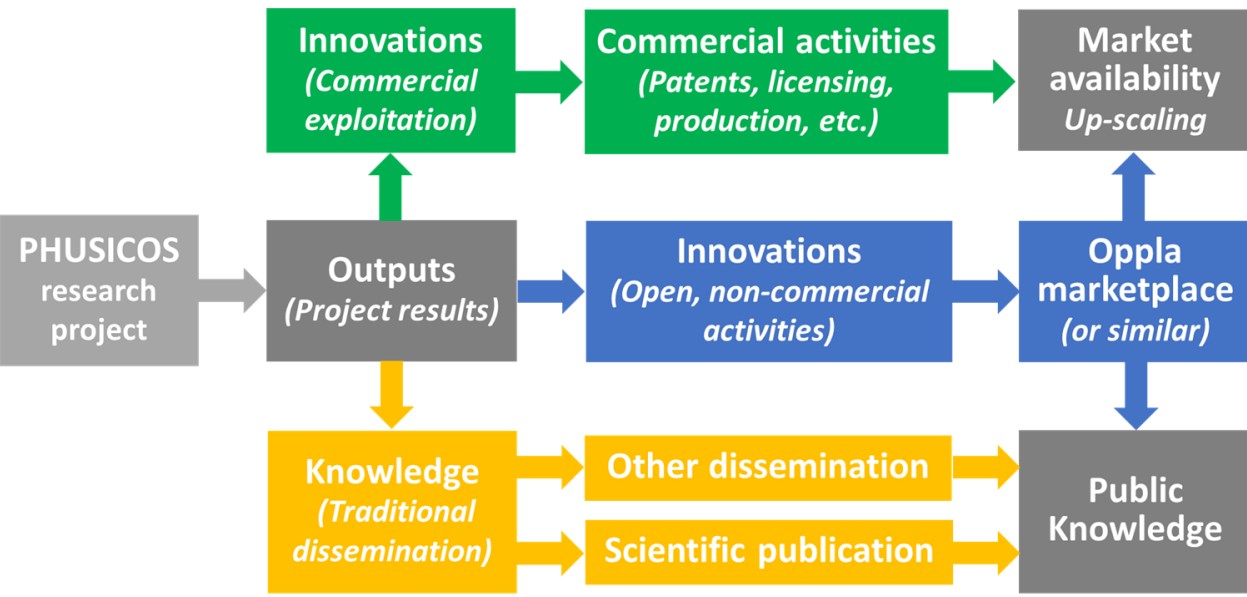

**Figure 3.** Pathways for the innovation process in PHUSICOS (illustration by author).

### 2.2. A Practical Example: The PHUSICOS Assessment Framework Tool

An example of an early PHUSICOS result is the assessment framework (AF) tool for evaluating NBSs, which was developed within the project's WP for Technical Innovation (WP4) and described in a deliverable from this WP [9]. The tool is based on selecting performance indicators relevant for a specific NBS and performing a multi-criteria decision analysis to score the NBS. The tool contains several novel results, including a comprehensive generalized set of performance indicators grouped by criteria and ambits, and a multi-level weighting methodology for scoring indicators, criteria, and ambits according to a bottom-up approach (Figure 4).

The innovative aspects of the AF tool are its scalability and adaptability to different territorial contexts and to sector-specific analyses, as well as its ability to be applied for either ex ante or ex post assessments. The former provides for the selection of the most suitable design scenario among a set of available ones, whereas the latter allows for monitoring of the effectiveness of the implemented scenario against the pre-intervention one (baseline scenario). Moreover, greater attention is given to social and economic impacts compared to existing frameworks. The living labs methodology [10] will be used to identify relevant performance indicators.

The AF tool is a technical innovation, and will be tested, customized, and refined through its application to both the demonstrator sites and the concept cases. Further developments to this tool to support innovation and up-scaling could include developing practical application tools, for example spreadsheet-based utilities for scoring and establishing relative assessments of NBS implementations or as an assessment at various stages within a specific NBS.

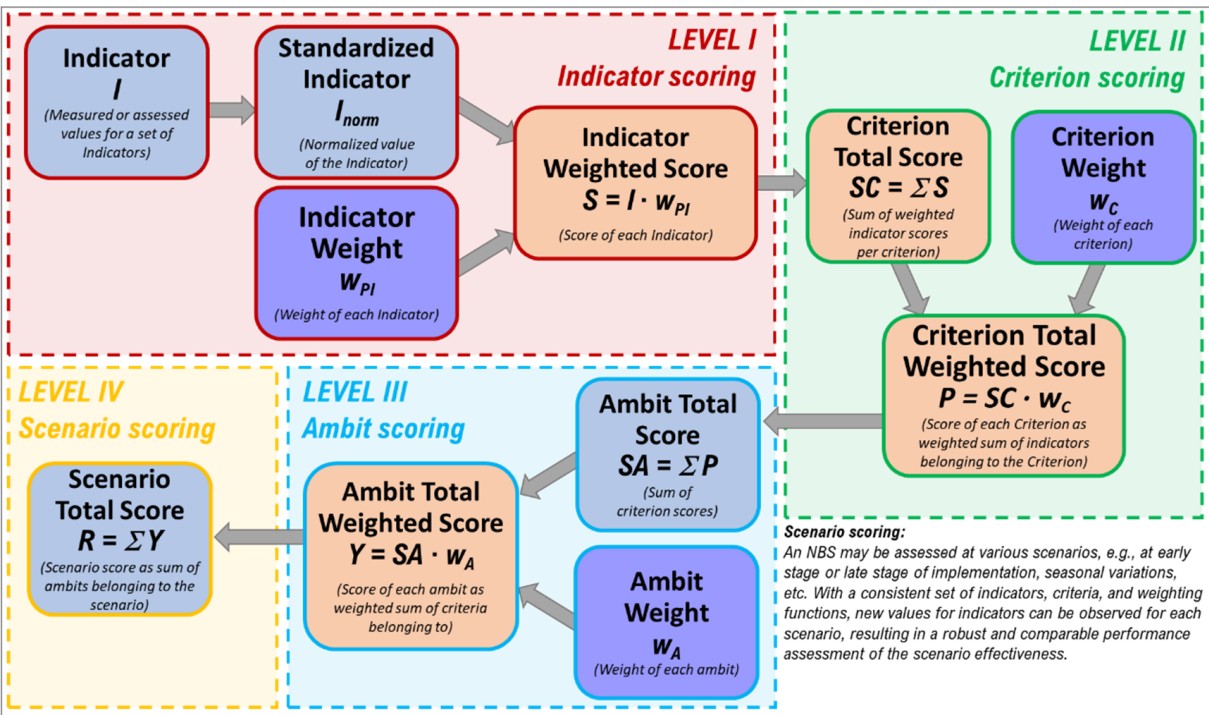

**Figure 4.** Assessment framework tool-scoring flow chart (illustration by author, adapted from [9]).

### 2.3. Making Research Results Accessible for Innovation

The PHUSICOS project consortium has established that the primary focus for innovation within the project will be open access results or open innovation, although this does not preclude potential commercial processes if the partner owning the IP sees this as relevant. The initial channel to the market for open innovation will be via information repositories or public databases, for example the Oppla Marketplace [11], European Open Science Cloud [12], and the OpenAIRE initiative [13]. Results that have been identified as innovations and intended for dissemination through these public databases need to be prepared for further utilization, dissemination, and up-scaling, e.g., making the results accessible for a larger group of stakeholders.

In PHUSICOS, this essentially consists of "packaging" the research results together with other essential information and structuring this in a way that a target user will have the information needed to successfully use the results. For example, consider the AF tool. The information needed to make this tool accessible for others may include:

- The PHUSICOS deliverable describing the method (a document);
- The table indicating the full matrix of ambits, criteria, and indicator parameters (a spreadsheet);
- An example using one of the PHUSICOS demonstrator sites (a spreadsheet showing the reduced matrix specific for the site, with weightings and calculations as practical examples);
- A list of key parameters suitable for identifying the content and purpose of this result, for example language, license terms (e.g., open source), and contact details. These metadata will be used to help identify the result when it is made available on various databases, for example Oppla Marketplace. (Oppla is the EU repository of nature-based solutions, which is accessible via the Oppla internet portal [11].)

These items would define a complete "product package" for the AF tool that could be posted in the public databases.

### 2.4. Up-Scaling of Innovations

Up-scaling takes a niche innovation, known only to a few and with little specific tangible value, to something that is accessible, available, familiar, and can be broadly applied, creating value for the parties involved in its implementation and use (Figure 5).

An up-scaling strategy should be anchored within a viable business model, addressing several key aspects of this, including the value creation potential (value proposition), who the stakeholders are, what their interests or needs are, and how the innovation can be made broadly available.

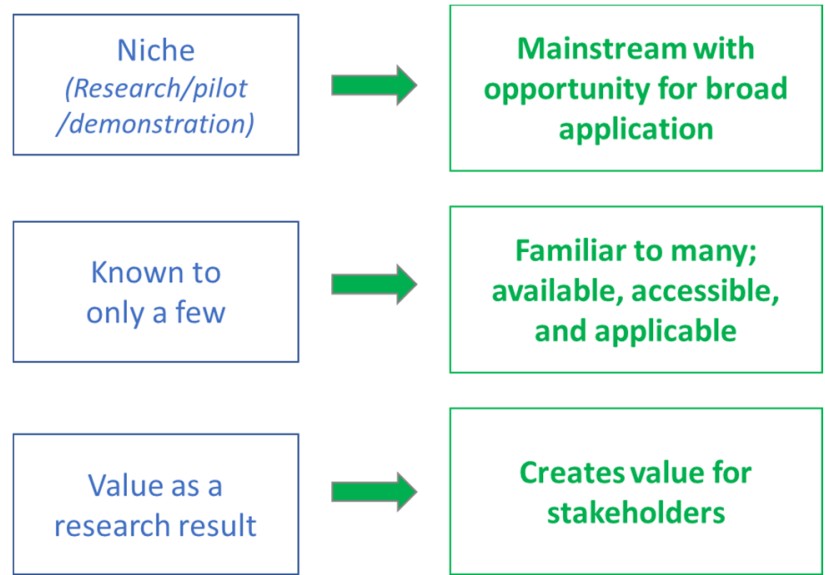

**Figure 5.** Up-scaling (illustration by author).

We can define the reach of our innovations using a nested scale (local, regional, national, and European), and define the approach for up-scaling as:

- Literal: Expanding the reach of a product, service, or knowledge to increasingly wider extents, e.g., up-scaling from a niche solution at one location to something that is well known and can be easily implemented at any location across Europe;
- Figurative: Generalizing the nature of a product, service, or knowledge, such that it becomes relevant and valuable for larger contexts, e.g., from being a local skill to becoming a European standard or recommended practice.

In some cases, up-scaling may be straightforward and quick, but unfortunately it will more often be a complicated and time-consuming process relying on factors, elements, or actors outside of the project's control. For example, incorporating a design method into a national or international standard may require years of committee work. Therefore, it is important to define appropriate sub-goals (milestones) that are achievable within the project timeline that positively contribute to the overall vision for up-scaling of the innovation after the research project is completed.

### 2.5. Performance Indicators for Up-Scaling

Tangible results and innovations will be made available for public use or provided as commercial services to the market at the different scales. Ideally, the success in meeting this goal would be directly measured via selected indicator parameters measuring the uptake of PHUSICOS innovations in other projects, activities, and contexts, e.g., up-scaling. Indicator parameters could be defined as measurable entities, for example the number of NBS designed and constructed or the number of countries implementing policy documents. However, this uptake will take time, and it will most likely be difficult or it will be unfeasible to measure this effect quantifiably within the timeframe of the PHUSICOS project. The

alternative is to identify indicators relevant for enabling up-scaling, dissemination, and uptake of the PHUSICOS results. The key assumption of course is that if these indicators are positive, they signal that the process of up-scaling is started and can continue after PHUSICOS is completed.

An analogy may be drawn from journal publications—key performance indicators for researchers are the number of journal articles published and the relative standing (rating) of the journals. However, there may be a significant delay from when the article is initially submitted to when it finally appears in "print", as this depends on peer review, revisions, etc. As alternative indicators, researchers may use "accepted for publication", "abstract submitted", "under peer review", or other nomenclature to specify the intention to publish and the overall progress of the publication process.

Essentially, up-scaling in PHUSICOS will be enabled by identifying innovations, making these accessible (see Section 2.6), and by "marketing" the innovations via social media or non-scientific fora and encouraging the partners owning the results to publish these in traditional technical or scientific fora (journals, conferences, etc.).

As an indicator for enabling up-scaling, the project will count the number of individual innovations made openly accessible and their subsequent promotion on social media. Specifically, the number of:

- Innovations published on Oppla Marketplace or similar (minimum 1 per work package);
- Postings on social media (minimum 3 mentions per published innovation).

Result owners will be encouraged to produce additional publications describing their innovations or presenting applications of their innovations as a means of increased technical dissemination.

### 2.6. Structuring and Tracking Innovation Up-Scaling within the PHUSICOS Project

EU research projects often involve multiple participants collaborating on a complex set of tasks and activities. In PHUSICOS, complexity is added by the core focus being on co-creation with stakeholders for the implementation of the NBS demonstrator projects. With this mix of stakeholders, it is difficult to maintain the focus on the overreaching goals of innovation through implementation and up-scaling. There is a clear need to develop a method or tool for identifying, structuring, and tracking innovation processes.

A tracking schema addressing relevant topics was developed in PHUSICOS. The purpose of this schema is two-fold: first, it will provide a convenient tool for the project management team to follow-up innovation processes; second, the process of filling in the schema encourages researchers to reflect on their contributions and activities in the project in the specific context of innovation. The schema is populated with specific detailed information for each WP. The information is specified both as overarching information for the WP, but also specified for each scale for up-scaling activities. An example is given in Table 2. Individual topics in the figure include:

- Innovation expectations: Products, services, and knowledge expected from the WP;
- Stakeholder interests: The relevant stakeholders and their specific interests;
- Up-scaling needs: Specified for local, regional, national, and European levels;
- Potentials for value creation: Describe how the products will create value for various stakeholders. Consider both economic value and non-economic benefits;
- Up-scaling needs: Which specific actions or measures are required to achieve effective up-scaling at each of the four scales;
- Challenges: Potential pitfalls, difficulties, or obstacles that may prevent up-scaling;
- Timeframe: Reasonable estimates for time required to achieve up-scaling;
- Actions and sub-goals for up-scaling: Specific targets or goals to be achieved within the project timeframe that will help ensure that up-scaling continues after the contractual end of the project.

**Table 2.** Innovation tracking schema for PHUSICOS WP 4 (Technical Innovations).

| | | | | |
|---|---|---|---|---|
| **Innovation expectations** | Innovations developed in WP4 may include: <br>• A comprehensive framework for NBS assessment <br>• Database/platform for monitoring and early warning <br>• Methods for developing hazard and risk maps to illustrate flood patterns and landslides for different climate scenarios <br>• Methods for evaluating ecosystems and ecosystem services for alternative landscape scenarios with plan designs | | | |
| **Stakeholder interests** | The most relevant stakeholder groups for these innovations are "Experts" and "Authorities": Experts as users of these tools, and Authorities as beneficiaries of the knowledge/information created. | | | |
| **Scale:** | Local | Regional | National | European |
| **Potentials for value creation** | Tools supporting development and monitoring of NBS and hybrid solutions, improving ecosystems. | Tools for managing risk and protecting environment. Flood and landslides are addressed in PHUSICOS, however other hazards can be added over time to reach a broader application base. | | Input at policy level for embracing NBS in risk reduction and ecosystem management across Europe. |
| **Upscaling needs** | Dissemination of the technical innovations, and development of an evidence base supporting use of these innovations for other cases. Innovative methods need to be clearly described in recommended practice documents appropriate for the target audience. Business opportunities may exist for software applications, for example digital toolboxes for geographic information systems (GIS) or worksheet templates implementing analysis routines. (literal upscaling) | | | Successful demonstration cases using the technical innovations, providing documentation for lobbyists to use when promoting NBS for disaster risk reduction (DRR). |
| **Challenges** | Uncertain if this tool will be useable on a local scale, as the broad technical and scientific competency needed may not be available in small communities | The tools and methods from this work package can clearly be distributed via Oppla Marketplace and other fora to make them available across larger scales (regional to European). Creating a commercial business case for products, for example software applications, may require investing time in developing interest and support in special interest groups. However, software applications aimed at the European market may face challenges related to national laws and requirements. | | |
| **Time scale** | Dissemination via Oppla Marketplace can be implemented immediately. Commercial developments, for example a software product, will require a longer period to develop a business case, secure investors and develop the product. | | | |
| **PHUSICOS actions and sub-goals** | • Create concise summary of the following results and publish these on the Oppla Marketplace: <br>→ the assessment framework tool including the supporting worksheets <br>→ monitoring data from the early warning database <br>→ methodology for risk map production <br>→ methodology for ecosystem services assessment <br>• Create informational announcements and release on appropriate social media | | | |

## 3. Summary

This technical note outlines some of the key concepts and principles employed within PHUSICOS for managing and promoting innovation processes within the project. A key aspect of this is recognizing the value of incremental innovations as stepping stones to showcase (major) innovations, opening up the identification of individual innovations within all work packages. The consortium prioritizes open innovation and open access; project results and innovations will be disseminated through public repositories and databases to encourage broad implementation and up-scaling.

The coordination and management of innovation activities involves a simple schematic approach to structure and organize specific innovation process considerations at various scales (local, regional, national, and European):

- Stakeholder interests and needs;
- Potential for value creation;
- Up-scaling needs;
- Specific challenges;
- Timeframe needed for up-scaling;
- Specific actions needed to implement the up-scaling

Up-scaling of the innovations will likely require more time than the overall PHUSICOS project duration. To address this, the PHUSICOS consortium will use indirect performance indicators to measure innovations. These include the dissemination of results, tools, and supporting documentation into the public repositories and databases (ensure availability) and promotion of awareness of these materials via social media channels and through traditional scientific dissemination.

**Author Contributions:** Conceptualization, J.M.S.; Investigation, J.M.S., F.P. and S.B.; Writing—original draft, J.M.S.; Project administration, A.M.P.O.; Funding acquisition, A.M.P.O.; Supervision, A.M.P.O.; Methodology; B.G.K., A.S.; Writing—review and editing, A.S., F.P. and G.L.; All authors have read and agreed to the published version of the manuscript.

**Funding:** PHUSICOS has received funding from the European Union's Horizon 2020 research and innovation program under grant agreement No. 776681.

**Institutional Review Board Statement:** Not applicable.

**Informed Consent Statement:** Not Applicable.

**Conflicts of Interest:** The authors declare no conflict of interest.

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
