# Peer review of "Innovation in NBS Co-Design and Implementation"

_sustainability, doi:10.3390/su13020986_

Round 1

Reviewer 1 Report

This paper outlines the main concepts of the PHUSICOS project and describes the NBS-based innovation. There are no major problems with the paper. However, there are some things that the author could modify to make this paper even better.

In the introduction, the authors describe PHUSICOS, but are there any similar projects out there? It is thought that the good points of PHUSICOS can be emphasized more by comparing it with other projects. Therefore, it is recommended to add some literature on the history in previous projects and projects in other countries to highlight the new aspects in PHUSICOS.

We believe that this AF tool has a very important place in PHUSICOS, because the new results from the AF tool for NBS evaluation are only in the references and the good points of the AF tool are not shown. By explaining the results of this AF tool in a more concrete way, we believe that we can provide the readers with a good idea of this AF tool and also a good idea of the PHUSICOS project.

It is very important to consider the up scaling of innovation. In this paper, there are shown the importance, indicators and concepts of upscaling. However, while these concepts are well understood, I wonder if there are some examples provided. I think that providing examples would provide useful information to readers.

Author Response

Please see the attached document which provides a complete reply to all comments provided.  The attached manuscript is also revised with 'track changes'.

Reviewer 2 Report

In my opinion, the article is not suitable for publication in the Sustainable journal. The article is descriptive. It describes the PHUSICOS project, which implements innovative activities under the H2020 program. It aims to demonstrate the use of natural solutions to mitigate hydrometeorological hazards in rural and mountain areas. Why, in my opinion, the article should not be published:

  1. No methodology, no scientific research.
  2. The article is not of a research nature, but rather of a review, descriptive, informative character.
  3. Since this is a review article (I understand that the authors want to inform many potential readers about PHUSICOS), it should be a "review article" or "short communication" about the project's goals and possibilities.
  4. Since this is a review article where there is no scientific research (because there is no research, no methodology, no research object or results), why is there such a large number of authors?

Author Response

(The authors gave the same response as above.)

Reviewer 3 Report

Dear Autors!

There are far too few literature references (only 10 items, the other 3 are websites).
In lines 51, 58, 122 there is terror, it is not known what exactly the author had in mind
Figures 3 and 5 appear to be illegible
Figure 1.2 seems superfluous
In the summary there is no indication of the applied test method according to the template.
Information about it in point 41 : 1.2. What is an innovation? is not very developmental. Of course, you can make an introduction about innovations, but you do not need to write a separate point. In my subjective opinion it is too obvious to create a separate point.
Immediately after the summary there are individual points 1.1 and 1.2 I would resign from it because in my subjective opinion they disturb the construction of the article.
At the end of the introduction, there is no description of the aim of the article, the research methodology used (e.g. also the research hypothesis put forward).
Some elements from the introduction should be placed in the middle of the study (e.g. Table 1. Stakeholder groups).
Figure 4: This is nothing new, it is unnecessary. There are no references in Figures 3 and 4.
There is also no description under the figures.
The key tab of the research methods used is omitted. There is no discussion and the results are immediately followed up with conclusions.
There is a lack of clear statements of the authors, conclusions of their opinions.
Figure 7 should be in discussion, not in conclusion. It looks like a table thrown in without any explanation etc.
I guess that if this program (PHUSICOS received funding from the EU research and innovation under the Horizon 2020 program on the basis of the grant agreement No. 776681) received funding, it probably must have developed a research methodology, but the authors did not disclose it. Can they not disclose it? Have they forgotten to add information about it?
In my opinion, there is a lack of quantitative data in international terms.
The article is very interesting, some thought processes are presented in brief. I think that it is very developing, but there is a lack of support for the quantitative data and the research methodology used. I suggest increasing the value of the article by at least the applied research methodology and (not necessarily) research hypotheses and remodeling as suggested.

Lots of Health. Yours sincerely

Reviewer

Author Response

(The authors gave the same response as above.)

Round 2

Reviewer 2 Report

The Authors' answers in the cover letter to my questions and doubts from the previous review have been cleared. I have no more comments.